# T6SS Accessory Proteins, Including DUF2169 Domain-Containing Protein and Pentapeptide Repeats Protein, Contribute to Bacterial Virulence in T6SS Group_5 of *Burkholderia glumae* BGR1

**DOI:** 10.3390/plants11010034

**Published:** 2021-12-23

**Authors:** Namgyu Kim, Gil Han, Hyejung Jung, Hyun-Hee Lee, Jungwook Park, Young-Su Seo

**Affiliations:** 1Department of Integrated Biological Science, Pusan National University, Busan 46241, Korea; titanic622@pusan.ac.kr (N.K.); croone@pusan.ac.kr (G.H.); jhj4059@pusan.ac.kr (H.J.); ehyuna92@pusan.ac.kr (H.-H.L.); 2Environmental Microbiology Research Team, Nakdonggang National Institute of Biological, Resources (NNIBR), Sangju 37242, Korea; jjuwoogi@nnibr.re.kr

**Keywords:** type VI secretion system, T6SS accessory proteins, T6SS adaptor, DUF2169 domain, pentapeptide repeats-containing protein

## Abstract

*Burkholderia glumae* are bacteria pathogenic to rice plants that cause a disease called bacterial panicle blight (BPB) in rice panicles. BPB, induced by *B. glumae*, causes enormous economic losses to the rice agricultural industry. *B. glumae* also causes bacterial disease in other crops because it has various virulence factors, such as toxins, proteases, lipases, extracellular polysaccharides, bacterial motility, and bacterial secretion systems. In particular, *B. glumae* BGR1 harbors type VI secretion system (T6SS) with functionally distinct roles: the prokaryotic targeting system and the eukaryotic targeting system. The functional activity of T6SS requires 13 core components and T6SS accessory proteins, such as adapters containing DUF2169, DUF4123, and DUF1795 domains. There are two genes, *bglu_1g23320* and *bglu_2g07420*, encoding the DUF2169 domain-containing protein in the genome of *B. glumae* BGR1. *bglu_2g07420* belongs to the gene cluster of T6SS group_5 in *B. glumae* BGR1, whereas *bglu_1g23320* does not belong to any T6SS gene cluster in *B. glumae* BGR1. T6SS group_5 of *B. glumae* BGR1 is involved in bacterial virulence in rice plants. The DUF2169 domain-containing protein with a single domain can function by itself; however, *Δu1g23320* showed no attenuated virulence in rice plants. In contrast, *Δu2g07420DUF2169* and *Δu2g07420PPR* did exhibit attenuated virulence in rice plants. These results suggest that the pentapeptide repeats region of the C-terminal additional domain, as well as the DUF2169 domain, is required for complete functioning of the DUF2169 domain-containing protein encoded by *bglu_2g07420*. *bglu_2g07410*, which encodes the pentapeptide repeats protein, composed of only the pentapeptide repeats region, is located downstream of *bglu_2g07420*. *Δu2g07410* also shows attenuated virulence in rice plants. This finding suggests that the pentapeptide repeats protein, encoded by *bglu_2g07410*, is involved in bacterial virulence. This study is the first report that the DUF2169 domain-containing protein and pentapeptide repeats protein are involved in bacterial virulence to the rice plants as T6SS accessory proteins, encoded in the gene cluster of the T6SS group_5.

## 1. Introduction

*Burkholderia glumae* was first reported in Japan in the 1950s [1]. *B. glumae* are bacteria pathogenic to rice plants, which cause a disease called bacterial panicle blight (BPB) in rice panicles [2,3]. *B. glumae* has several virulence factors, such as toxins, proteases, lipases, extracellular polysaccharides, bacterial motility, and bacterial secretion systems [4,5,6,7,8,9,10]. Thus, *B. glumae* not only cause BPB in rice panicles, but also bacterial wilt in rice stems and various other economically important crops, such as tomatoes, eggplants, hot peppers, and sesame [3,4]. The virulence factors of *B. glumae* contribute to its resistance against the host immune system and cause damage the host. In particular, bacterial secretion systems have specialized, sophisticated, and highly developed strategies for the selective transport of toxic proteins (as effector proteins) across bacterial cell membranes. *B. glumae* BGR1 has four functionally distinct type VI secretion systems (T6SS): T6SS group_1, T6SS group_2, group_4, and T6SS group_5. T6SS group_1 of *B. glumae* BGR1 is involved in bacterial competition, whereas T6SS group_4 and group_5 contribute to bacterial virulence [3]. However, the type VI secreted proteins of *B. glumae* BGR1 have not clearly been identified.

One of the bacterial secretion systems, the type VI secretion system (T6SS), is a contractile nanomachine which is prevalent in 25% of Gram-negative bacteria and contributes to interspecies interaction in a contact-dependent manner with adjacent bacteria or eukaryotic cells [11,12,13]. T6SS is composed of 13 major components (TssA–TssM), assorted into specific structures, i.e., a cell-envelope-associated membrane complex, a baseplate complex, and a needle complex [14,15,16]. In addition to these major components, several accessory proteins related to T6SS, called T6SS accessory proteins, work with T6SS-dependent effectors to perform their functions.

A cell-envelope-associated membrane complex is assembled using TssM, TssL, and TssJ. The baseplate complex is assembled from TssE, TssF, and TssG, and the baseplate complex and membrane complex are connected by TssK. TssA is involved in baseplate assembly and TssB/TssC in sheath polymerization. The needle complex comprises TssD as an inner tube, a TssB/TssC sheath surrounding the inner tube, and a VgrG/PAAR spike complex at its end [17,18]. The VgrG/PAAR protein spike complex is formed of the VgrG trimer with PAAR and is responsible for creating an opening in the target cells [19,20,21]. The VgrG/PAAR protein spike complex acts on its own as an effector if it contains an additional C-terminal toxin domain [20]. Another effector, called the cargo effector, is bound through non-covalent interactions with structural components, such as VgrG, PAAR protein, and TssD, either directly, or with the aid of an adapter protein or chaperone [22,23,24,25]. A cargo effector is considered to be one of several T6SS-associated proteins. In particular, effectors that are supported by adaptor/chaperones, which are not secreted by the T6SS, are indispensable for mounting on the spike complex and for enabling secretion from bacterial cells [26].

T6SS-dependent effectors are injected directly into adjacent target cells or with the aid of T6SS accessory proteins, such as T6SS adaptor/chaperones [25,26]. T6SS adaptors/chaperones include specific protein domains, such as DUF1795, DUF4123, and DUF2169 domains. In addition, T6SS adaptor/chaperones interact with T6SS-dependent effectors and VgrG (TssI) based on protein–protein interactions. The chaperone/adaptor with the DUF4123 domain was identified through co-immunoprecipitation and bacterial two-hybrid analysis in *V**ibrio cholerae* through its interactions with VgrG-1 and T6SS-dependent effectors (TseL) [27]. DUF4123 domain-containing proteins of *V. cholerae* are required for the secretion of TseL and TseL-mediated antibacterial activity [28]. However, the identity and function of T6SS-dependent effectors are still not well understood.

In this study, DUF2169 domain-containing protein and pentapeptide repeats protein, which is involved in the bacterial virulence activity of *B. glumae* BGR1, was identified. The genes of *bglu_2g07420* and *bglu_2g07410*, the functionally unidentified genes encoding these two proteins, are present in the gene cluster of T6SS group_5 in *B glumae* BGR1. Among them, *bglu_2g07420* encodes the region of the DUF2169 domain, known as T6SS adapter/chaperones at its N-terminus and the region of pentapeptide repeats at its C-terminus. In addition, *bglu_1g23320*, which encodes only the DUF2169 domain in the genome of *B. glumae* BGR1, does not belong to the four T6SS gene clusters in *B. glumae* BGR1. Only loss of the DUF2169 domain of *bglu_2g07420* belonging to the gene cluster of T6SS group_5 showed attenuated virulence in rice plants. In addition, the bacterial virulence of *B. glumae* BGR1 was found to be affected by the loss of the pentapeptide repeats region in *bglu_2g07420* and *bglu_2g07410*. *bglu_2g07410* encodes the pentapeptide repeats protein with only the pentapeptide repeats region. Our findings show that *bglu_2g07420* and *bglu_2g07410*, belonging to the gene cluster of T6SS group_5 in *B. glumae* BGR1, encode the DUF2169 domain-containing protein and pentapeptide repeats protein, respectively, and contribute to the bacterial virulence to rice plants as the T6SS-associated proteins.

## 2. Results

### 2.1. Investigation and Analysis of the Genes Containing the DUF2169 Domain and Its Adjacent Gene in the Genome of Burkholderia glumae BGR1

Some T6SS accessory proteins, such as DUF1795, DUF4123, and DUF2169 domain-containing proteins, are known T6SS adapters that require the loading of a specific effector onto the cognate VgrG for delivery [25]. The DUF2169 domain-containing protein was found to be a T6SS accessory protein in plant pathogenic bacteria, *A. tumefaciens*; therefore, we focused on only the genes encoding the DUF2169 domain of the T6SS accessory protein [22]. The only two genes encoding the DUF2169 domain in the genome of *B. glumae* BGR1, *bglu_1g23320* and *bglu_2g07420*, were identified using the National Center for Biotechnology Information (https://www.ncbi.nlm.nih.gov/protein; accessed on 17 August 2021). In particular, *bglu_2g07420* was present in the gene cluster of T6SS group_5 in *B. glumae* BGR1 (Figure 1A). However, *bglu_1g23320* was not included in the gene cluster of any T6SS group and existed independently as a separate entity (Figure 1B). The *bglu_1g23320* gene is a hypothetical protein (WP_015876299.1) composed of 362 amino acids, whereas *bglu_2g07420* is a hypothetical protein (WP_012733555.1) composed of 878 amino acids. Functional analysis of the proteins encoded by the two genes was performed using the InterPro database (http://www.ebi.ac.uk/interpro; accessed on 17 August 2021) (Table 1). The *bglu_1g23320* gene encodes only the DUF2169 domain, whereas the *bglu_2g07420* gene encodes the DUF2169 domain at the N-terminus and the pentapeptide repeats region at the C-terminus (Figure 2). Additionally, the non-annotated gene, *bglu_2g07410*, located adjacent to *bglu_2g07420*, encoding the DUF2169 domain-containing protein, was also analyzed with the InterPro database. The *bglu_2g07410* gene, which is located downstream of the *bglu_2g07420* gene, was evaluated as a pentapeptide repeats protein (WP_012733554.1), encoding only the pentapeptide repeats region consisting of 356 amino acids (Figure 1A and Table 1). We assumed that the *bglu_2g07420* gene, which is present in the gene cluster of T6SS group_5, and the other gene of *bglu_1g23320*, which is not present in any T6SS group, are the major putative T6SS accessory proteins involved in the function of T6SS group_5 in *B. glumae* BGR1. Furthermore, we assumed that the *bglu_2g07410* gene encoding the pentapeptide repeats region, similarly to the *bglu_2g07420* gene with its pentapeptide repeats region, is involved in the function of T6SS group_5 as a putative T6SS accessory protein.

### 2.2. Construction of Markerless Deletion Mutants to Evaluate Whether the Putative T6SS Accessory Protein Affects the Function of T6SS Group_5 in B. glumae BGR1

To investigate the functions of putative T6SS accessory proteins containing the DUF2169 domain, we constructed markerless deletion mutants targeting the DUF2169 domain of *bglu_1g23320* and *bglu_2g07420* genes via homologous recombination. Single markerless deletion mutants, called *Δu2g07420DUF2169* and *Δu1g23320*, were generated via the deletion of only the DUF2169 domain of *bglu_2g07420* and *bglu_1g23320*, respectively (Appendix A). To complement the phenotypes of *Δu2g07420DUF2169* and *Δu1g23320*, the respective complementation strains, *Δu2g07420DUF2169*-C and *Δu1g23320*-C, were generated.

To investigate whether the pentapeptide repeats region encoded in *bglu_2g07410* and C-terminus of *bglu_2g07420* affect the function of T6SS group-5 as putative T6SS accessory proteins, the pentapeptide repeats region of both genes was deleted to construct markerless deletion mutants. Single markerless deletion mutants, called *Δu2g07420PPR* and *Δu2g07410*, were generated by deleting the pentapeptide repeats region of *bglu_2g07420* and *bglu_2g07410*, respectively (Appendix A). To complement the phenotypes of *Δu2g07420PPR* and *Δu2g07410*, the respective complementation strains, *Δu2g07420PPR*-C and *Δu2g07420*-C, were generated. A double markerless deletion mutant, *Δu2g07410-20*, was generated to determine whether two separate pentapeptide repeats regions independently contributed to the function of T6SS group_5. To complement the phenotype of *Δu2g07410-20*, the respective complementation strain, *Δu2g07410-20*-C, was generated.

### 2.3. The DUF2169 Domain Encoded by the N-terminus of bglu_2g07420 inside the Gene Cluster of T6SS Group_5 Is Involved in Bacterial Virulence in Rice Plants

To determine whether the DUF2169 domain encoded by *bglu_1g23320* and *bglu_2g07420* is involved in the functional role of T6SS group_5 in bacterial virulence, causing bacterial panicle blight, rice plants at the flowering stage were infected with wild-type *B. glumae* BGR1 and the mutants *Δu1g23320* and *Δu2g07420DUF2169* (Figure 3). At 8 days post-inoculation (dpi), most of the wild-type BGR1-inoculated panicles had developed blight disease symptoms (severity, 0–5), with a disease severity score of 4.38 ± 0.079 for wild-type BGR1 (Figure 3B). In contrast, *Δu2g07420DUF2169* exhibited attenuated virulence, with a disease severity score of 2.73 ± 0.134 (Figure 3B). The restored strain, *2g07420DUF169*-C, completely recovered its virulence, with a disease severity score of 4.27 ± 0.26 (Figure 3B). Rice plants were also infected with the *Δu1g23320* mutant to determine whether the *bglu_1g23320* gene contributed to bacterial virulence. However, the disease severity score of *Δu1g23320* was 4.26 ± 0.046 (Figure 3B). The deletion of the DUF2169 domain in *bglu_1g23320* caused almost no difference in pathogenicity from wild-type BGR1, although the deletion of the DUF2169 domain in *bglu_2g07420* differed from that of wild-type BGR1 (Figure 3). Thus, only the DUF2169 domain (WP_012733555.1), which is encoded by *bglu_2g07420* in the gene cluster of T6SS group_5, is involved in bacterial virulence to rice plants.

### 2.4. The Pentapeptide Repeats Region Encoded by bglu_2g07410 and the C-terminus of bglu_2g07420 inside the Gene Cluster of T6SS Group_5 Is Involved in Bacterial Virulence to Rice Plants

To determine whether only the DUF2169 domain encoded by the N-terminus of *bglu_2g07420* is involved in the functional role of T6SS group_5, rice plants at the flowering stage were infected with *Δu2g07420PPR*, in which the C-terminal pentapeptide repeats region is deleted. At 8 dpi, most of the wild-type BGR1-inoculated panicles had developed blight disease symptoms (severity, 0–5), with a disease severity score of 4.38 ± 0.079 for wild-type BGR1 (Figure 4B). In contrast, *Δu2g07420PPR* displayed attenuated virulence, with a disease severity score of 3.05 ± 0.022 (Figure 4B). To determine whether the pentapeptide repeats region is also involved in bacterial virulence, rice plants were infected with *Δu2g07410*, in which *bglu_2g07410* was deleted downstream of *bglu_2g07420*. *Δu2g07410* also showed attenuated virulence, with a disease severity score of 2.59 ± 0.114 (Figure 4B). *Δu2g07410-20* exhibited attenuated virulence, with a disease severity score of 1.85 ± 0.091 (Figure 4B), as did *Δu2g07410-20* (Figure 4B). The restored strains, namely, *Δu2g07420PPR*-C, *Δu2g07410*-C, and *Δu2g07410-20*-C, completely recovered their virulence, with disease severity scores of 4.32 ± 0.279, 4.26 ± 0.193, and 4.18 ± 0.179, respectively (Figure 4). The levels of attenuated virulence by single mutants, *Δu2g07420* and *Δu2g07410*, were similar at the reproductive and flowering stages of rice plants (Figure 4 and Figure 5). Thus, the pentapeptide repeats regions of *bglu_2g07420* and *bglu_2g07410* also contribute to bacterial virulence in rice plants. However, the results of in vivo pathogenicity assays at the reproductive stage showed that the bacterial virulence of *Δu2g07410-20*, in which both genes were deleted, was more attenuated than that of single mutants (Figure 4). The mean disease severity score of *Δu2g07410-20* and the mean disease severity scores of *Δu2g07410* or *Δu2g07420* were compared using the least significant difference at *p* < 0.001 according to Tukey’s HSD test.

## 3. Discussion

T6SS has evolved into the most complex secretory pathway because of its key role during interactions with the surroundings [29]. T6SS is known to have two distinct functional features: a prokaryotic targeting system with antibacterial effects manifesting as bacterial interactions, and a eukaryotic targeting system demonstrating bacterial virulence as eukaryotic interactions. T6SS is a proteinaceous machine with 13 core components. The core genes of T6SS are organized into genomic clusters. In addition to the core components, the gene cluster of T6SS additionally encodes accessory genes functionally associated with T6SS [20,30,31,32,33,34]. These accessory proteins are either involved in the secretion of T6SS-dependent effectors or regulate the assembly and activity of T6SS [35]. The functions of most T6SS accessory proteins have not been identified. In particular, the protein comprising the DUF2169 domain exists as a single-domain protein with only the DUF2169 domain; however, some are multi-domain proteins with additional domains as small portions at the C-terminus [36]. The most common additional domain of the DUF2169 domain-containing protein is the pentapeptide repeats region, which has unknown functions [36].

The DUF2169 domain-containing protein of a single-domain protein can be involved in the functional role of T6SS as an adapter, even if there is no additional domain at the C-terminus. For example, a single DUF2169 domain-containing protein encoded by *atu3641* in *Agrobacterium tumefaciens* C58 is a T6SS adapter associated with Tde2, an effector of T6SS, and contributes to the secretion of Tde2 [22].

*B. glumae* BGR1 has a DUF2169 domain-containing protein encoded by two genes, *bglu_1g23320* and *bglu_2g07420*, on its genome. These genes are located downstream of the *vgrG*. The genes which encode T6SS accessory proteins, such as adaptors, are often found downstream of *vgrG* [27]. *bglu_1g23320* encodes a single-domain protein with only the DUF2169 domain. The *bglu_1g23320* gene was not included in any T6SS gene cluster of *B. glumae* BGR1 (Figure 1). Some T6SS accessory genes and its upstream gene, *vgrG*, are located outside the T6SS gene cluster and borrow the core components of T6SS [37]. However, *Δu1g23320* did not show attenuated virulence in rice plants (Figure 3). T6SS adapters, such as the DUF2169 domain-containing protein, enable secretion by loading specific effectors into cognate VgrG (Valine glycine repeat G) without affecting the function of T6SS [22,27,35]. Therefore, the DUF2169 domain-containing protein encoded by *bglu_1g23320*, which is not included in any T6SS gene cluster of *B. glumae* BGR1, is not involved in a specific effector that directly causes pathogenicity in rice plants, or may remain as a non-functional T6SS accessory protein.

In contrast to *bglu_1g23320*, *bglu_2g07420* encodes a multi-domain protein with the DUF2169 domain and pentapeptide repeats region as an additional domain at the C-terminus. Additionally, *bglu_2g07420* is included in the T6SS group_5 gene cluster of *B. glumae* BGR1. *bglu_2g07420* encodes a DUF2169 domain-containing protein and is present in the gene cluster of T6SS group_5; therefore, a DUF2169 domain-containing protein encoded by *bglu_2g07420* can be considered as a T6SS accessory protein of T6SS group_5 in *B. glumae* BGR1. The T6SS group_5 of *B. glumae* BGR1 is involved in bacterial virulence in rice plants [3]. In addition, the T6SS-5 gene cluster of *B. pseudomallei* and *B. thailandensis* belonging to the eukaryotic system has *tagA/B-5*, which encodes the DUF2169 domain-containing protein [38]. Additionally, TagA/B-5 of *B. thailandensis* is a multi-domain protein composed of the DUF2169 domain and pentapeptide repeats region as an additional domain at the C-terminus, which is essential for full virulence in multinucleated giant cells [39]. Thus, the DUF2169 domain-containing protein encoded by *bglu_2g07420* was also hypothesized to be essential for bacterial virulence as a T6SS accessory protein of the T6SS group_5 in *B. glumae* BGR1.

The single-domain protein containing DUF2169 can function by itself; therefore, *Δu2g07420DUF2169* was generated by deleting only DUF2169 of the DUF2169 domain-containing protein without causing any problem in the frameshift of *bglu_2g07420*. Furthermore, an in vivo pathogenicity assay was performed with this strain. The attenuated virulence of *Δu2g07420DUF2169* indicated that the DUF2169 domain of the DUF2169 domain-containing protein encoded by *bglu_2g07420* is required for bacterial virulence in rice plants. *Δu2g07420PPR* was generated to determine whether the DUF2169 domain-containing protein encoded by *bglu_2g07420* can fully function as a T6SS accessory protein even in the absence of an additional domain. An in vivo pathogenicity assay was also performed using this strain. Similarly to *Δu2g07420DUF2169*, the attenuated virulence of *Δu2g07420PPR* indicated that the DUF2169 domain-containing protein encoded by *bglu_2g07420* requires not only the DUF2169 domain, but also the additional C-terminal domain, the pentapeptide repeats region, to enable bacterial virulence in rice plants. Therefore, the DUF2169 domain-containing protein encoded by *bglu_2g07420* is a functional T6SS accessory protein consisting of multiple domains having the DUF2169 domain and an additional C-terminal pentapeptide repeats region.

*bglu_2g07410*, located downstream of *bglu_2g07420* which encodes the additional C-terminal domain of the pentapeptide repeats region, encodes only the pentapeptide repeats region. The protein containing only the pentapeptide repeats region might also contribute to bacterial virulence in rice plants with the DUF2169 domain-containing protein having an additional C-terminal pentapeptide repeats region. Indeed, *Δu2g07410* also showed attenuated virulence in rice plants. The attenuated virulence of *Δu2g07410* indicates that the pentapeptide repeats protein, encoded by *bglu_2g07410*, also contributes to bacterial virulence in rice plants. Pentapeptide repeats proteins are widely distributed in both prokaryotes and eukaryotes [40]. These proteins may be responsible for targeting, or for structural functions, rather than enzymatic activity [40]. These proteins have a superhelical structure and can participate in protein–protein interactions without conferring biological functions [40]. For example, the pentapeptide repeats regions of MfpA and Qnr have a superhelical structure, mimicking the B-form of DNA and inhibiting DNA gyrase, conferring resistance to fluoroquinolone [41,42,43,44]. However, the functions or roles of pentapeptide repeats proteins are not fully understood.

Interestingly, the type III secretion system-dependent effector, PipB2, in *Salmonella* spp. contains the pentapeptide repeats region, which directly interacts with kinesin-1 and contributes to the formation of *Salmonella*-induced filaments. However, the exact function and role of the pentapeptide repeats region in PipB2 is also unclear [41,42,43]. Interestingly, in the results of the in vivo pathogenicity assay at the reproductive stage, the severely attenuated virulence by *Δu2g07410-20* was shown to be a synergistic effect of the two T6SS accessory proteins encoded by *bglu_2g07410* and *bglu_2g07420*. *B. glumae* BGR1 mainly causes BPB as a serious symptom in the reproductive stage of rice plants, but can also cause diseases at any developmental stage, such as seed rot, stunting in seedlings, and sheath rot in the vegetative stage. *Neopseudocercosporella capsellae*, a plant pathogen, causes disease in canola, and the incidence and the severity of the disease are determined by the growth stage of the host plant in which this pathogen invasion occurs [45]. Therefore, the results of the in vivo pathogenicity assay at the reproductive stage show that *B. glumae* BGR1 induces BPB, with distinct functions of these two T6SS accessory proteins involved in bacterial virulence at the reproductive stage.

In the current study, we obtained genes encoding the DUF2169 domain from the genome of *B. gluma*e BGR1 and investigated whether the genes encoding the DUF2169 domain-containing proteins consisting of a single domain or multi-domain affect bacterial virulence to rice plants. Of the two genes encoding the DUF2169 domain-containing protein, *bglu_1g23320* and *bglu_2g07420*, only the latter was involved in bacterial virulence. We discovered two genes encoding the T6SS accessory proteins in the gene cluster of T6SS group_5 in *B. glumae* BGR1: *bglu_2g07420* and *bglu_2g07410*. Our results show that the DUF2169 domain-containing protein encoded in *bglu_2g074720* is a multi-domain protein that requires both the DUF2169 domain and the pentapeptide repeats region to contribute to bacterial virulence. T6SS accessory protein, having a DUF2169 domain, is a well-known T6SS adaptor. The DUF2169 domain-containing protein encoded by *bglu_2g07420* could be considered as a potential T6SS adaptor; however, additional experimental evidence, such as interactions with VgrG or specific T6SS-dependent effectors, is required to confirm its function as a T6SS adaptor. Furthermore, we showed that another T6SS accessory protein, encoded by *bglu_2g07410* located downstream of *bglu_2g07420*, is involved in bacterial virulence as the pentapeptide repeats protein. To the best of our knowledge, this is the first report on the identification of the T6SS accessory proteins, the DUF2169 domain-containing protein, and the pentapeptide repeats protein, belonging to T6SS group_5 of *B. glumae* BGR1 and involved in bacterial virulence. Novel T6SS accessory proteins involved in the bacterial virulence of *B. glumae* BGR1 can be usefully employed to understand the functional activity of T6SS.

## 4. Materials and Methods

### 4.1. Bacterial Strains, Plasmids, and Growth Conditions

All bacterial strains and plasmids used in this study are detailed in Table 2. The wild-type BGR1, mutant strains, and *E. coli* were cultured on Luria Bertani (LB) agar and in broth medium at 37 °C with shaking at 200 rpm. The growth curve of bacterial strains was determined every 2 h by measuring the optical density at 600 nm (OD_600_) using a UV-1800 spectrophotometer (Shimadzu, Kyoto, Japan). When required, LB media were supplemented with antibiotics at concentrations of 100 μg/mL (rifampicin) and 50 μg/mL (kanamycin).

### 4.2. Construction of Markerless Deletion Mutants and Their Complemetation Strains in B. glumae BGR1

All primer pairs used for the construction of deletion mutants are shown in Appendix A. Plasmids, including pK18*u1g23320*, pK18*u2g07410*, pK18*u2g07420DUF2169*, and pk18*u2g07420PPR*, harboring portions of the target gene upstream (L fragment) and downstream (R fragment), were prepared. The L and R fragments were designed according to the Gibson assembly reaction. The L and R fragments were ligated by secondary PCR using LF and RR primer pairs. To construct pK18*u1g23320*, the ligated LR fragments and pK18*mobsacB* were digested with BamHI-HindIII (NEB, Ipswich, MA, USA). To construct pK18*u2g07410*, the ligated LR fragments and pK18*mobsacB* were digested with EcoRI and HindIII. To construct pK18*u2g07420DUF2169*, the ligated LR fragments and pK18*mobsacB* were digested with EcoRI and HindIII. To construct pK18*u2g07420PPR*, the ligated LR fragments and pK18*mobsacB* were digested with BamHI and HindIII. Competent *E. coli* DH5a cells were transformed with recombinant plasmids and selected in kanamycin-containing media. Recombinant plasmids were transformed into *E. coli* S17-1 cells. Recombinant plasmids were transformed into *B. glumae* BGR1 by conjugation through co-culture with *E. coli* S17-1 on LB agar plates. The first crossover was selected on media containing kanamycin (60 μg/mL) and rifampicin (100 μg/mL). The first crossover of colonies was confirmed by PCR using the primer pairs of genes, _UP_F and pK18_DOWN_R. The selected cells underwent a second crossover in LB broth containing kanamycin (60 μg/mL). The second crossover in the cells was selected on LB agar, containing 30% sucrose and rifampicin (100 μg/mL). Finally, single deletions of *bglu_1g23320*, *bglu_2g07410*, the DUF2169 domain region, and the pentapeptide repeats region of *bglu_2g07420* gene were confirmed by PCR using the primer sets for gene_UP_F and gene_DOWN_R. The double deletion mutant, *Δ2g07410-20*, was constructed based on single *Δ2g07410* deletion mutants using the above-mentioned methods.

For complementation, the entire open reading frame of *bglu_2g07410* and *bglu_2g07420* genes was amplified by PCR using the primer pairs Cgene_F and Cgene_R. The amplified DNA fragments were cloned into the broad-host-range expression vector, pBBR1MCS2. The cloned vectors, pB*u2g07410* and/or pB*u2g07420*, were conjugated into single and double deletion mutant strains by co-culturing with S17-1. Finally, complementation strains of single and double deletion mutants were selected on media containing rifampicin (100 μg/mL) and kanamycin (60 μg/mL) and were confirmed by PCR using primer pairs of pB_UP_F and pB_DOWN_R.

### 4.3. In Vivo Pathogenicity Assay at the Vegetative and Reproductive Stages of Rice Plants

To examine the bacterial virulence of putative T6SS accessory proteins in *B. glumae* BGR1 at the vegetative and reproductive stages of rice plants, cultured bacterial cells in the mid-logarithmic phase were harvested by centrifugation, washed, and resuspended in distilled water. Thereafter, the optical density of the bacterial suspensions was adjusted to OD_600nm_ = 0.8. The rice plants (*Oryza sativa* L. Saeilmi) used in this experiment were grown under greenhouse conditions (average 30 °C in the day, and 25 °C at night). Subsequently, the stems of vegetative-stage rice plants were inoculated with bacterial suspensions using a syringe and grown for 8 days. Disease severity was observed in the inoculated area. Suspensions for each bacterial strain were prepared and adjusted to OD_600nm_ = 0.5 to confirm bacterial panicle blight disease at the rice reproductive stage. At the flowering stage, the rice panicles were inoculated via dipping into 50 mL bacterial suspensions for 1 min. At 8 dpi, the disease severity of the rice panicles in individual rice plants was evaluated using the following scale: 0—healthy panicle; 1—0–20% diseased panicle; 2—20–40% diseased panicle; 3—40–60% diseased panicle; 4—40–80% diseased panicle; 5—80–100% diseased panicle. Disease severity was calculated using the following formula: disease severity = Σ (number of samples per rating × rating value)/total number of panicles.

### 4.4. Statistical Analysis

All experiments were conducted at least three times, with at least three replicates. Analyses of the disease severity scores were conducted with at least three replicates of individual rice plants. The mean disease severity scores between two groups were compared using the least significant difference test according to Tukey’s HSD assay.

## Figures and Tables

**Figure 1 plants-11-00034-f001:**
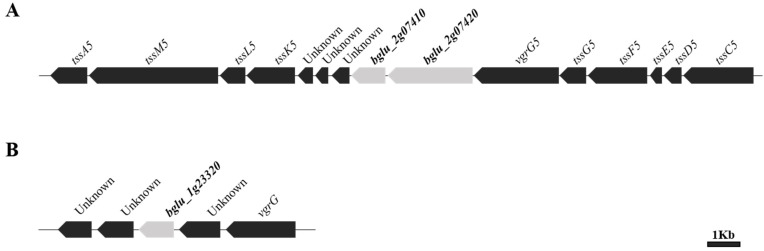
Genetic organization of T6SS group_5, containing *bglu_2g07420* encoding the DUF2169-containing protein and *bglu_2g07410* encoding pentapeptide repeats protein (**A**), and *bglu_1g23320* encoding the DUF2169-containing protein (**B**) in *Burkholderia glumae* BGR1. The genes are indicated by the locus ID (e.g., *bglu_2g07410*) and are to scale. The genes marked in gray encode the DUF2169 domain-containing proteins and pentapeptide repeats protein.

**Figure 2 plants-11-00034-f002:**
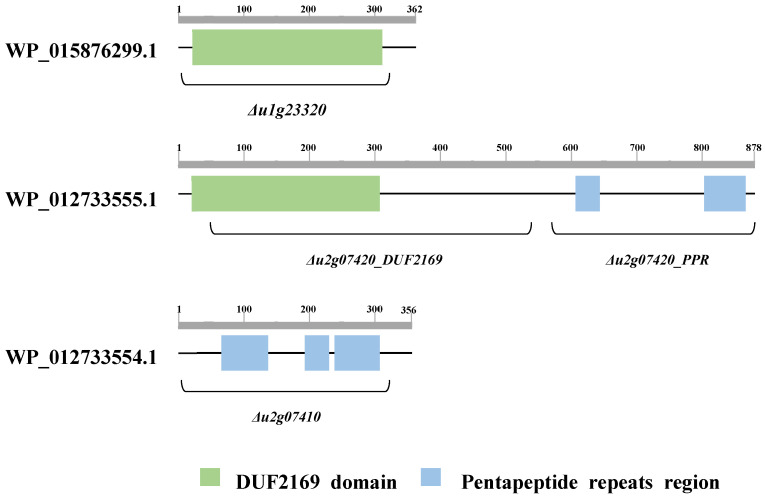
Domain architectures of WP_015876299.1, WP_012733554.1, and WP_012733555.1. Only proteins with distinct domain organizations are shown. The DUF2169 domain-containing protein (WP_015876299.1) encoded by *bglu_1g23320* has the DUF2169 domain as a single-domain protein. The DUF2169 domain-containing protein (WP_012733555.1) encoded by *bglu_2g07420* has multiple domains with pentapeptide repeats region at the C-terminus. The pentapeptide repeats protein (WP_012733554.1) encoded by *bglu_2g07410* only has the pentapeptide repeats region. The deletion sites of the mutant strains used in this study are shown with *Δu1g23320*, *Δu2g07420DUF2169*, *Δu2g07420PPR*, and *Δu2g07410*.

**Figure 3 plants-11-00034-f003:**
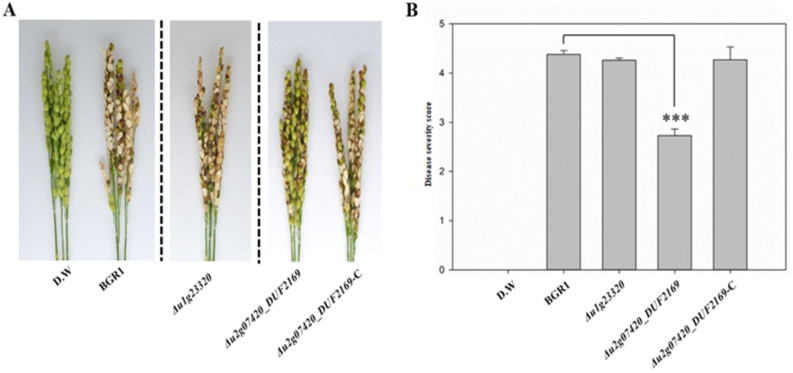
In vivo pathogenicity assay at the reproductive stage to assess the virulence of *Δu1g23320* and *Δu2g07420DUF2169*. Representative of three replicates (**A**). Disease severity on the rice panicles was calculated on a scale of 0 to 5 after inoculating the bacterial suspension (**B**). Data are presented as the mean ± S.D. of three replicates (*n* = 3). Mean values followed by the same letters are not significantly different according to Tukey’s HSD test (*** *p* < 0.001). Disease symptoms at 8 days post-inoculation. Distilled water (D.W.) was used as the negative control.

**Figure 4 plants-11-00034-f004:**
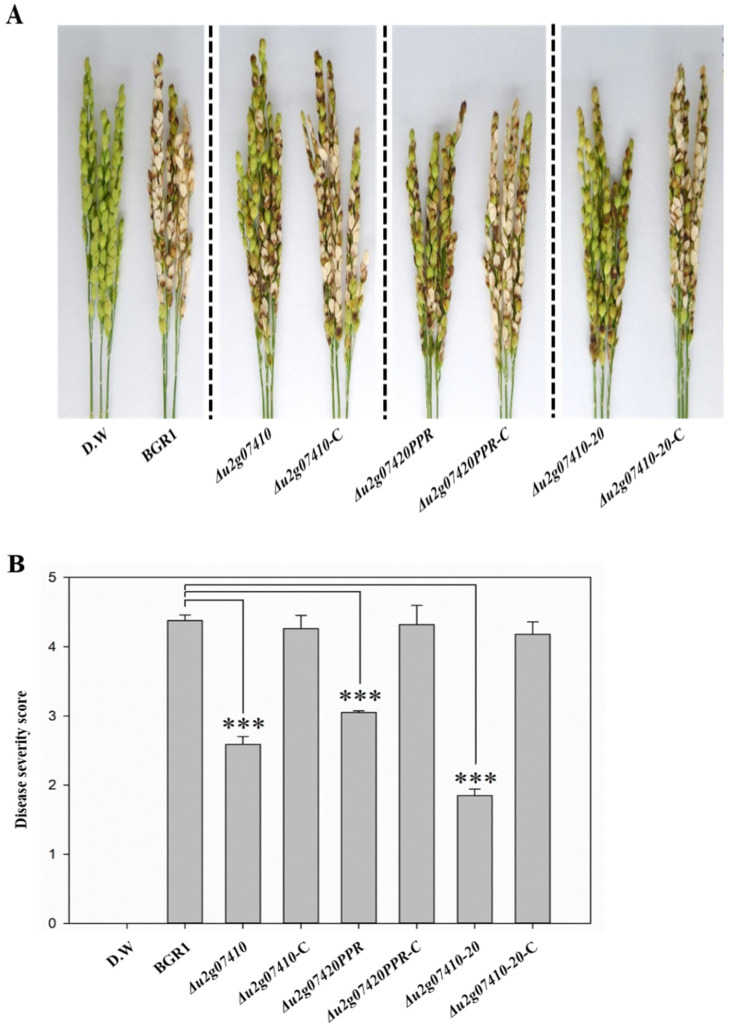
In vivo pathogenicity assay at the reproductive stage to assess the virulence of *Δu2g07410*, *Δu2g07420PPR*, and *Δu2g07410-20*. Representative of three replicates (**A**). Disease severity on the rice panicles was calculated on a scale of 0 to 5 after inoculating the bacterial suspension (**B**). Data are presented as the mean ± S.D. of three replicates (*n* = 3). Mean values followed by the same letters are not significantly different according to Tukey’s HSD test (*** *p* < 0.001). Disease symptoms at 8 days post-inoculation. Distilled water (D.W) was used as the negative control.

**Figure 5 plants-11-00034-f005:**
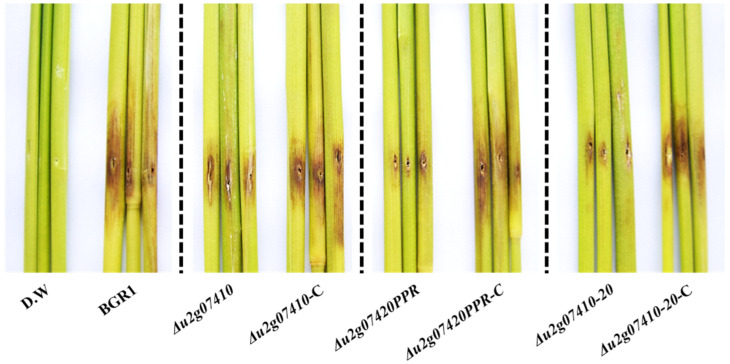
In vivo pathogenicity assay at the vegetative stage to assess the virulence of *Δu2g07410*, *Δu2g07420PPR*, and *Δu2g07410-20*. Wild-type and mutant strains were inoculated in the stems of rice plants. Representative of three replicates. Disease symptoms at 8 days post-inoculation. Distilled water (D.W) was used as the negative control.

**Table 1 plants-11-00034-t001:** Functional analysis of WP_015876299.1, WP_012733554.1, and WP_012733555.1 from the InterPro database.

Analysis	Accession	Description	InterPro Accession	InterPro Description	Interval	E-Value
WP_015876299.1						
Pfam	PF09937	Uncharacterized protein conserved in bacteria (DUF2169)	IPR018683	DUF2169	21~311	5.6 × 10^−91^
WP_012733554.1						
Pfam	PF13599	Pentapeptide repeats (9 copies)	IPR001646	Pentapeptide repeat	66~137	2.5 × 10^−10^
Pfam	PF00805	Pentapeptide repeats (8 copies)	IPR001646	Pentapeptide repeat	193~230	3.2 × 10^−9^
Pfam	PF13599	Pentapeptide repeats (9 copies)	IPR001646	Pentapeptide repeat	238~308	1.7 × 10^−7^
WP_012733555.1						
Pfam	PF09937	Uncharacterized protein conserved in bacteria (DUF2169)	IPR018683	DUF2169	20~307	6.4 × 10^−73^
Pfam	PF00808	Pentapeptide repeats (8 copies)	IPR001646	Pentapeptide repeat	605~642	2.0 × 10^−11^
Pfam	PF13599	Pentapeptide repeats (9 copies)	IPR001646	Pentapeptide repeat	801~864	3.1 × 10^−6^

**Table 2 plants-11-00034-t002:** Bacterial strains and plasmids used in this study.

Name	Characteristics	Source
**Bacterial strains**		
** *B. glumae* **		
BGR1	*Burkholderia glumae* isolate from rice, wild-type, Rif^r^ *	[4]
*Δu2g07410*	BGR1 derivative, deletion of 824 bp within *bglu_2g07410*	This study
*Δu2g07420_DUF2169*	BGR1 derivative, deletion of 1466 bp of the DUF2169 domain region within *bglu_2g07420*	This study
*Δu2g07420_PPR*	BGR1 derivative, deletion of 927 bp of the pentapeptide repeats region within *bglu_2g0720*	This study
*Δu2g07410-20*	*bglu_2g07410* and pentapeptide repeats region of *bglu_2g07420* double deletion mutant derived from BGR1	This study
*Δu1g23320*	BGR1 derivative, deletion of 951 bp within *bglu_1g23320*	This study
*Δu2g07410*-C	BGR1 *bglu_2g07410* deletion mutant containing pB*u2g07410*	This study
*Δu2g07420_DUF2169*-C	BGR1 *bglu_2g07420* deletion mutant (*Δu2g07420_DUF2169*) containing pB*u2g07420*	This study
*Δu2g07420_PPR*-C	BGR1 *bglu_2g07420* deletion mutant (*Δu2g07420_PPR*) containing pB*u2g07420*	This study
*Δu2g07410-20*-C	BGR1 *bglu_2g07410-20* double deletion mutant (*Δu2g07410-20*) containing pB*u2g07410* and pB*u2g07420*	This study
** *E. coli* **		
*E. coli* DH5α λpir	F^-^ 80d*lacZ*ΔM15 (*lacZYA-argF*) U169 *recA1 endA1hsdR17* (rk-, mk+) *phoAsupE44* -*thi-1 gyrA96 relA1*	Lab collection
*E. coli* S17-1 λpir	*hsdR recA* pro RP4-2 (Tc::Mu; Km::Tn7) (*λ pir*)	[46]
**Plasmids**		
pK18*mobsacB*	Allelic exchange suicide vector, *sacB* Km^r^ *	[47]
pK18*u2g07410*	For constructing bglu_2g07410 KO mutant, pK18mobsacB:: LR fragment of *bglu_2g07410* region restricted by EcoRI-HindIII	This study
pK18*u2g07420DUF2169*	For constructing the DUF2169 domain in the *bglu_2g0742*0 KO mutant, pK18*mobsacB*:: LR fragment of the DUF2169 domain region in bglu_2g07420 restricted by EcoRI-HindIII	This study
pK18*u2g07420PPR*	For constructing pentapeptide repeats region in the *bglu_2g07420* KO mutant, pK18mobsacB::LR fragment of pentapeptide repeats region in bglu_2g07420 restricted by BamHI-HindIII	This study
pK18*u1g23320*	For constructing *bglu_1g23320* KO mutant, pK18*mobsacB*:: LR fragment of *bglu_1g23320* region restricted by BamHI-HindIII	This study
pBBR1MCS2	Broad-host-range plasmid, Km^r^ *, used to construct complementation strains.	[48]
pB*u2g07410*	For constructing the *Δu2g07410* complementation strain, pBBR1MCS2::CDS of *bglu_2g07410*	This study
pB*u2g07420*	For constructing the *Δu2g07420_DUF2169* and *Δu2g07420_PPR* complementation strain, pBBR1MCS2::CDS of *bglu_2g07420*	This study

* Abbreviations: Rif^r^—rifampicin resistance; Km^r^—kanamycin resistance.

## Data Availability

Data sharing is not applicable to this article.

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
