# Peer review of "T6SS Accessory Proteins, Including DUF2169 Domain-Containing Protein and Pentapeptide Repeats Protein, Contribute to Bacterial Virulence in T6SS Group_5 of *Burkholderia glumae* BGR1"

_plants, 2021, doi:10.3390/plants11010034_

Round 1
Reviewer 1 Report
I checked your manuscript and described comments below.
The important point of this paper is "this is the first report on the identifi-32 cation of DUF2169 domain-containing protein and pentapeptide repeat region-containing protein, 33 which are novel T6SS accessory proteins that support the function of T6SS group_5 of B. glumae 34 BGR 1 ".
The following comments are minor problem.
- If possible, you should add following NCBI accession ID about protein.
bglu_2g07410: WP_012733554.1
bglu_2g07420: WP_012733555.1
bglu_1g23320: WP_015876299.1
- If you are using a program for Tukey ’s HSD test, you should write it in detail.
- You must change reference 12 from Science (80-) to Science.
Author Response
We thank the reviewers for the valuable comments and suggestions for improvement. We enclose our revised manuscript entitled " T6SS accessory proteins, including DUF2169 domain-containing protein and pentapeptide repeats protein, contribute to bacterial virulence in T6SS group_5 of Burkholderia glumae BGR1". Below, we address point by point the issue raised, and indicated the changes made in the accompanying revised manuscript. All changes in the revised manuscript are marked in red.
Reviewer 1
The important point of this paper is "this is the first report on the identification of DUF2169 domain-containing protein and pentapeptide repeat region-containing protein, which are novel T6SS accessory proteins that support the function of T6SS group_5 of B. glumae BGR 1 ".
The following comments are minor problem.
Point 1: If possible, you should add following NCBI accession ID about protein.
bglu_2g07410: WP_012733554.1
bglu_2g07420: WP_012733555.1
bglu_1g23320: WP_015876299.1
Response 1: We have revised and modified the NCBI accession ID about protein in the Table 1 and Figure 2.
Point 2: If you are using a program for Tukey ’s HSD test, you should write it in detail.
Response 2: We have revised and modified the following the statement in section of Statistical analysis: “All experiments were conducted at least twice, with at least three replicates. Analysis of the disease severity score were conducted by at least three replicates of individual rice plants. The mean value of disease severity score between two groups were compared using the least significant difference test according to the Tukey's HSD test (Line 402-405).”
Point 3: You must change reference 12 from Science (80-) to Science.
Response 3: We have revised the reference 11 from Science (80-) to Science (Line 438).

Reviewer 2 Report
Major comments:
In this manuscript, the authors investigated the functions of the DUF2169-containing proteins of Burkholderia glumae BGR1. The virulence of the targeting gene-deficient mutations was evaluated and compared with that of strain BGR1 through in vivo pathogenicity assay. They found that the DUF2169 protein encoded by bglu_2g07420 and the pentapeptide repeats protein encoded by bglu_2g07410 contributed to bacterial virulence. Although the experimental approaches are straightforward, there are several major concerns on the data presented in the manuscript, which require more careful interpretation. The manuscript should be improved.
Line 19-21: Since it is known that the DUF2169 protein encoded by bglu_2g07420 relates to bacterial virulence, the pathogenic study on DUF2169 domains does not have much more significance.
Figure 1, table 1 and figure 2: The result from pentapeptide repeats protein is interesting. Therefore, the information of bglu_2g07410 should be analyzed and presented in figure 1, figure 2 and table 1. The discussion at line 286-298 mentioned that the pentapeptide repeat region-containing protein has a superhelical structure and can participate in protein-protein interactions without conferring biological functions. It is more convincing to further investigate how the pentapeptide repeats protein affects the virulence of BGR1.
Figure 3, 4, 5: These pathogenicity assays should be conducted separately for three times with three repeats to analyze the statistical significance. In addition, the number of bacterial colonies per panicle should be measured after 8 days infection.
Line 300-305: The pathogenicity of Δu2g07410-20 looks like less attenuated than that of Δu2g07420PPR on rice stem, which is not consistent with the results on the rice panicles. The synergistic effect of two accessory proteins to bacterial virulence is not convincing.
Minor comments:
Table 3 might be provided as supplementary table.
Line 58: “comonents” is a spelling mistake.
Line 385: “except comparative proteomics analysis” may be deleted.
Author Response
We thank the reviewers for the valuable comments and suggestions for improvement. We enclose our revised manuscript entitled " T6SS accessory proteins, including DUF2169 domain-containing protein and pentapeptide repeats protein, contribute to bacterial virulence in T6SS group_5 of Burkholderia glumae BGR1". Below, we address point by point the issue raised, and indicated the changes made in the accompanying revised manuscript. All changes in the revised manuscript are marked in red.
Reviewer 2
Comments and Suggestions for Authors
Major comments:
In this manuscript, the authors investigated the functions of the DUF2169-containing proteins of Burkholderia glumae BGR1. The virulence of the targeting gene-deficient mutations was evaluated and compared with that of strain BGR1 through in vivo pathogenicity assay. They found that the DUF2169 protein encoded by bglu_2g07420 and the pentapeptide repeats protein encoded by bglu_2g07410 contributed to bacterial virulence. Although the experimental approaches are straightforward, there are several major concerns on the data presented in the manuscript, which require more careful interpretation. The manuscript should be improved.
Point 1: Line 19-21: Since it is known that the DUF2169 protein encoded by bglu_2g07420 relates to bacterial virulence, the pathogenic study on DUF2169 domains does not have much more significance. Figure 1, table 1 and figure 2: The result from pentapeptide repeats protein is interesting. Therefore, the information of bglu_2g07410 should be analyzed and presented in figure 1, figure 2 and table.
Response 1: We agree with the reviewer. We have mentioned and added the more information and presentation about pentapeptide repeats protein encoded by bglu_2g07410 in Figure 1, Figure 2 and Table 1.
Point 2: The discussion at line 286-298 mentioned that the pentapeptide repeat region-containing protein has a superhelical structure and can participate in protein-protein interactions without conferring biological functions. It is more convincing to further investigate how the pentapeptide repeats protein affects the virulence of BGR1.
Response 2: We made the following modification on the reviewer’s advice: “Pentapeptide repeats proteins are widely distributed in both prokaryotes and eukaryotes [40]. These proteins may be responsible for targeting or the structural functions rather than enzymatic activity [40]. These proteins have a superhelical structure and can participate in protein-protein interactions without conferring biological functions [40]. For example, the pentapeptide repeats region of MfpA and Qnr has a superhelical structure, mimicking the B-form of DNA and inhibiting DNA gyrase, conferring resistance to fluoroquinolone [41–44]. However, the function or role of pentapeptide repeats protein is not fully understood. Interestingly, the type III secretion system-dependent effector, PipB2, in Salmonella spp. contains the pentapeptide repeats region which directly interacts with kinesin-1 and con-tributes to the formation of Salmonella-induced filaments. However, the exact function and role of the pentapeptide repeats region in PipB2 is also unclear [41-43] (Line 300-310).”
Point 3: Figure 3, 4, 5: These pathogenicity assays should be conducted separately for three times with three repeats to analyze the statistical significance. In addition, the number of bacterial colonies per panicle should be measured after 8 days infection.
Response 3: All experiments were conducted at least three times, with at least three replicates. Analysis of the disease severity score were conducted by at least three replicates of individual rice plants. We have revised and added the analysis of the statistical significance (Line 401-404). The bacterial panicle blight (BPB) caused by B. glumae BGR1 was evaluated using the following scale: 0, healthy panicle; 1, 0-20% diseased panicle; 2, 20-40% dis-eased panicle; 3, 40-60% diseased panicle; 4, 40-80% diseased panicle; and 5, 80-100% diseased panicle. Disease severity was calculated using the following formula: disease severity = Σ (number of samples per rating × rating value) / total number of panicles. The pathogenicity of the rice pathogenic bacteria, B. glumae BGR1, was mainly evaluated by this formula without measuring bacterial colonies.
The following articles are some of examples.
Kim, et al., 2007. Regulation of polar flagellum genes is mediated by quorum sensing and FlhDC in Burkholderia glumae. Mol. Microbiol. 64:164-179.
Chun et al., 2009. The quorum sensing-dependent gene katG of Burkholderia glumae is important for protection from visible light. 191:4152-4157.
Chen et al. 2012. Dissection of quorum-sensing genes in Burkholderia glumae reveals non-canonical regulation and the new regulatory gene tofM for toxoflavin production. PLoS One. 7:e52150.
Goo et al., 2017. Lethal Consequences of Overcoming Metabolic Restrictions Imposed on a Cooperative Bacterial Population. mBio. 28,8:e00042-17.
Kim et al., 2020. Type VI secretion systems of plant‐pathogenic Burkholderia glumae BGR1 play a functionally distinct role in interspecies interactions and virulence. Molecular plant pathology. 21: 1055-1069.
Point 4: Line 300-305: The pathogenicity of Δu2g07410-20 looks like less attenuated than that of Δu2g07420PPR on rice stem, which is not consistent with the results on the rice panicles. The synergistic effect of two accessory proteins to bacterial virulence is not convincing.
Response 4: We modified this statement as follow: “The attenuated virulence by single mutants, which are Δu2g07420 and Δu2g07410, was similar at the reproductive stage of rice plants and flowering stage of rice plants (Figure 4 and 5). Thus, the pentapeptide repeats region of bglu_2g07420 and bglu_2g07410 also contribute to bacterial virulence in rice plants. However, the results of in vivo pathogenicity assay at reproductive stage showed that the bacterial virulence of Δu2g07410-20 in which both genes were deleted was more attenuated than that of single mutants (Figure 4) (Line 209-215).” “Interestingly, in the results of the in vivo pathogenicity assay at the reproductive stage, the severely attenuated virulence by Δu2g07410-20 was shown to be a synergistic effect of the two T6SS accessory proteins encoded by bglu_2g07410 and bglu_2g07420. B. glumae BGR1 mainly causes BPB as a serious symptom in the reproductive stage of rice plants, but can also cause diseases at any developmental stage, such as seed rot, stunting in seedlings, and sheath rot in the vegetative stage. Neopseudocercosporella capsellae, a plant pathogen, causes disease in canola, and disease incidence and severity are determined by the growth stage of the host plant in which this pathogen invasion occurs [45] (Line 310-318).” Thus, genes involved in pathogenicity in plant pathogenic Burkholderia species can show differential pathogenicity based on rice stage. Currently, we observed stage dependent pathogenicity in rice pathogenic Burkhoderia species and even B. glumae isolates.
Minor comments:
Point 1: Table 3 might be provided as supplementary table.
Response 1: We agree with the reviewer’s advice. Table 3 were changed to Supplementary Table 4.
Point 2: Line 58: “comonents” is a spelling mistake.
Response 2: We have revised the spelling from “comonents” to “components” (Line 57).
Point 3: Line 385: “except comparative proteomics analysis” may be deleted.
Response 3: We deleted “except comparative proteomics analysis” (Line 402).

Reviewer 3 Report
A few DUF2169-domain-containing proteins are known to be adaptor/accessory proteins necessary for bacterial virulence on plants via the function of type six secretion systems (T6SS).
In this study, the authors identify two predicted DUF2169-domain containing protein in the genome of Burkholderia glumae BGR1, the causal agent of rice bacterial panicle blight. One of these predicted proteins is located in the genomic context of BGR1 T6SS group 5 and is predicted to contain pentapeptide repeat regions (PPR, gene bglu_2g07420), while one is not in a T6SS genomic context and lacks identifiable a PPR domain (bglu_1g23320). Additionally, the authors identify a second gene (bglu_2g07410) just downstream of bglu_2g07420 which is predicted to encode only PPR and no DUF2169.
The authors produce mutant strains of BGR1 lacking each bglu_2g07420 and bglu_1g23320 and find that loss of the former gene reduces disease symptoms on rice, while mutation of the latter gene does not noticeably reduce disease symptoms. The authors also make deletion strains of bglu_2g07410 (i.e. the gene predicted to encode only PPR domain) and also observe reduced disease symptoms. In addition, the authors produce mutants lacking both 2g07420 and 2g07410 and report a synergistic effect of the double deletion. A BGR1 strain lacking just the PPR of 2g07420 also exhibits reduced ability to produce disease. Mutant BGR1 phenotypes are nicely genetically complemented, supporting the author's findings. The data are convincing and interesting.
I have a few easily addressed concerns about the presentation:
1. It is apparent that the two genes identified are important for rice disease symptoms caused by BGR1, and it is likely that the reduced virulence is because the encoded proteins (particularly the DUF2169/PPR-domain containing one) act as adaptors for a T6SS. However, the conclusions that these gene(s) are actual functional adaptors for T6SS would require further experimental data. For example, does the deletion mutant have a reduced capacity to deliver effectors? If these data are not presented, the conclusions must be toned down to indicate that the reduced virulence is likely but not conclusively the result of compromised T6SS function/effector delivery. Examples of this include the abstract (lines 34-35), line 95, lines 101-102, line 176, line 191, lines 267-268, lines 275-277, line 285, lines 305-306, line 308, line 313.
2. The presentation should include Sanger sequencing- or PCR-based data to show that the BGR1 deletion mutants are in fact deletion mutants. Supplementary data would be a fine place for it. (I apologize if the data are already present, but can see no links to Supp. material on the review website or in your manuscript.)
3. Are bglu_2g07410 and bglu_2g07420 actually two genes? They are immediately adjacent to each other so they could be misannotated as two genes when in fact they are a single expressed transcript (I have experience with a gene like this in a different organism). Can you present gene expression data to show that they are actually two separately transcribed genes? This could be accomplished by presenting gene expression data from a database somewhere, or by RT-PCR
4. Since you discuss bglu_2g07410 quite a bit, it should be included in Table 1 and Figure 2.
5. I would appreciate a figure which visually represents the locations of the deletions in the different deletion strains that you made.
6. In Figure 1A, please put gene names/IDs on the relevant discussed genes bglu_2g07410/2g07420 and bglu_1g23320.
Author Response
Reviewer 3
Point 1: It is apparent that the two genes identified are important for rice disease symptoms caused by BGR1, and it is likely that the reduced virulence is because the encoded proteins (particularly the DUF2169/PPR-domain containing one) act as adaptors for a T6SS. However, the conclusions that these gene(s) are actual functional adaptors for T6SS would require further experimental data. For example, does the deletion mutant have a reduced capacity to deliver effectors? If these data are not presented, the conclusions must be toned down to indicate that the reduced virulence is likely but not conclusively the result of compromised T6SS function/effector delivery. Examples of this include the abstract (lines 34-35), line 95, lines 101-102, line 176, line 191, lines 267-268, lines 275-277, line 285, lines 305-306, line 308, line 313.
Response 1: We made the following modifications based on the reviewer’s advice: “These results suggest that the pentapeptide repeats region of the C-terminal additional domain, as well as the DUF2169 domain, is required for the fully function of the DUF2169 do-main-containing protein encoded by bglu_2g07420. (Line 26-29)” “This finding suggests that the pentapeptide repeats protein encoded by bglu_2g07410 is involved in bacterial virulence. This study is the first report that DUF2169 domain-containing protein and pentapeptide repeats protein are involved in bacterial virulence to the rice plants as T6SS accessory protein, encoded in the gene cluster of T6SS group_5. (Line 31-35)” “DUF2169 domain-containing protein and pentapeptide repeats protein, which is involved in bacterial virulence of B. glumae BGR1, was identified. The genes of bglu_2g07420 and bglu_2g07410 as functionally unidentified genes encoding these two proteins are present in the gene cluster of T6SS group_5 in B glumae BGR1. (Line 87-90)” “Our findings show that bglu_2g07420 and bglu_2g07410, belonging to the gene cluster of T6SS group_5 in B. glumae BGR1, encode DUF2169 domain-containing protein and pentapeptide repeats protein, respectively, and that contribute to bacterial virulence to rice plants as the T6SS associated proteins. (Line 99-102)” “In the current study, we discovered the two genes, bglu_2g07420 and bglu_2g07410, encoding the T6SS accessory proteins in the gene cluster of T6SS group_5 in B. glumae BGR1. Our results show that the DUF2169 domain-containing protein, a multi-domain protein encoded by bglu_2g07420 and composed of DUF2169 domain and pentapeptide repeats region, contributes to the bacterial virulence in rice plants. T6SS accessory protein having DUF2169 domain is well known as a T6SS adaptor. The DUF2169 domain-containing protein encoded by bglu_2g07420 could be considered as a potential T6SS adaptor, but additional experimental evidence, such as interaction with VgrG or specific T6SS-dependent effectors is required to confirm its function as a T6SS adaptor. Furthermore, we showed that another T6SS accessory protein, encoded by bglu_2g07410 located in downstream of bglu_2g07420, is involved in bacterial virulence as the pentapeptide repeats protein. To our best knowledge, this is the first report on the identification of the T6SS accessory proteins, DUF2169 domain-containing protein and pentapeptide repeats protein, belonging to T6SS group_5 of B. glumae BGR1 and involved in bacterial virulence. The discovery of new T6SS accessory proteins involved in the bacterial virulence of B. glumae BGR1 can be usefully employed to understand the functional activity of T6SS. (Line 322-337)”
Point 2: The presentation should include Sanger sequencing- or PCR-based data to show that the BGR1 deletion mutants are in fact deletion mutants. Supplementary data would be a fine place for it. (I apologize if the data are already present, but can see no links to Supp. material on the review website or in your manuscript.)
Response 2: We agree with the reviewer’s advice. We have mentioned and added the Sanger sequencing data that the deletion mutant strains (Δu2g07410, Δu2g07420_DUF2169, Δu2g07420_PPR and Δu1g23320) in Supplementary Table 1, 2 and 3.
Point 3. Are bglu_2g07410 and bglu_2g07420 actually two genes? They are immediately adjacent to each other so they could be misannotated as two genes when in fact they are a single expressed transcript (I have experience with a gene like this in a different organism). Can you present gene expression data to show that they are actually two separately transcribed genes? This could be accomplished by presenting gene expression data from a database somewhere, or by RT-PCR
Response 3: Even if the two genes of bglu_2g07410 and bglu_2g07420 in the gene cluster of T6SS group_5 are directly adjacent to each other, bglu_2g07410 and bglu_2g07420 have different frameshifts. The fact that the two genes are in different frameshifts indicates that they are actually correctly annotated as the single gene.
Point 4: Since you discuss bglu_2g07410 quite a bit, it should be included in Table 1 and Figure 2.
Response 4: We agree with the reviewer. We have mentioned and added the more information and presentation about pentapeptide repeats protein encoded by bglu_2g07410 in Figure 1, Figure 2 and Table 1.
Point 5: I would appreciate a figure which visually represents the locations of the deletions in the different deletion strains that you made.
Response 5: We have added the location of the deletion site in deletion mutant strains in Figure 2, and Supplementary Table 1, 2, and 3.
Point 6: In Figure 1A, please put gene names/IDs on the relevant discussed genes bglu_2g07410/2g07420 and bglu_1g23320.
Response 6: We have revised about the relevant discussed genes about bglu_2g07410, bglu_2g07420 and bglu_1g23320. These genes are not yet annotated. We have mentioned gene IDs in Figure 1.

Reviewer 4 Report
Kim et al. revealed that bglu_2g07420 is involved in bacterial virulence as a one of the gene in the T6SS group_5 cluster, and both DUF2169 domain and pentapeptide repeat region are necessary for the virulence.
- Abstract can be improved. To consider the broader interest, the disease impact caused by B. glumae on rice would be put in the first sentence.
- The authors should explain the reason why they focus on bglu_1g23320 and bglu_2g07420, carrying DUF2169 domain in the first paragraph of Result section not in Discussion section (line 223- 234).
- In Fig.1, the name of 3 genes focused in this study should be labeled under diagrams as “bglu_2g07340”. Honestly, 4 gene IDs presented in Fig.1 seems not to be necessary.
- The definition of disease severity in Methods section needs more explanation. Does the percentage indicate “number of diseased panicles/total panicle number in one individual”? If so, how many individuals are used in this assay?
- The authors mentioned that “provided evidence that a specific DUF2169 domain-containing protein, consisting of multiple domains encoded in the T6SS group_5 gene cluster, is involved in the function of T6SS group_5 as a T6SS adapter.” in line 309 in the Discussion section. However, although they showed that truncated version of bglu_2g07420 attenuate its virulence, there are no molecular evidences that bglu_2g07420 works as a T6SS adapter. To advocate it, molecular experiments are needed such as pull-down assay to show that bglu_2g07420 makes complex with the other T6SS components.
Author Response
Reviewer 4
Comments and Suggestions for Authors
Kim et al. revealed that bglu_2g07420 is involved in bacterial virulence as a one of the gene in the T6SS group_5 cluster, and both DUF2169 domain and pentapeptide repeat region are necessary for the virulence.
Point 1: Abstract can be improved. To consider the broader interest, the disease impact caused by B. glumae on rice would be put in the first sentence.
Response 1: We agree with the reviewer. The Abstract has been revised in general (Line 12-35). The disease impact caused by B. glumae on rice have been mentioned in the first sentence of Abstract (Line 12-14).
Point 2: The authors should explain the reason why they focus on bglu_1g23320 and bglu_2g07420, carrying DUF2169 domain in the first paragraph of Result section not in Discussion section (line 223- 234).
Response 2: We made the following modification according to the reviewer’s advice: “Some T6SS accessory proteins, such as DUF1795, DUF4123, and DUF2169 domain-containing proteins, are known T6SS adapters that require the loading of a specific effector onto the cognate VgrG for delivery [25]. Among these T6SS accessory proteins, only two genes, (Line 110-113)”
Point 3: In Fig.1, the name of 3 genes focused in this study should be labeled under diagrams as “bglu_2g07340”. Honestly, 4 gene IDs presented in Fig.1 seems not to be necessary.
Response 3: We agree with the reviewer. We have revised and added the presentation about three genes, which are bglu_2g07410, bglu_2g07420, and bglu_1g23320, in Figure 1.
Point 4: The definition of disease severity in Methods section needs more explanation. Does the percentage indicate “number of diseased panicles/total panicle number in one individual”? If so, how many individuals are used in this assay?
Response 4: We have mentioned and added the following statement: “the disease severity of rice panicles in individual rice plant was evaluated (Line 396-397)” “All experiments were conducted at least twice, with at least three replicates. Analysis of the disease severity score were conducted by at least three replicates of individual rice plants. (Line 402-404)”
Point 5: The authors mentioned that “provided evidence that a specific DUF2169 domain-containing protein, consisting of multiple domains encoded in the T6SS group_5 gene cluster, is involved in the function of T6SS group_5 as a T6SS adapter.” in line 309 in the Discussion section. However, although they showed that truncated version of bglu_2g07420 attenuate its virulence, there are no molecular evidences that bglu_2g07420 works as a T6SS adapter. To advocate it, molecular experiments are needed such as pull-down assay to show that bglu_2g07420 makes complex with the other T6SS components.
Response 5: We modified this statement as follow: “Our results show that the DUF2169 domain-containing protein, a multi-domain protein encoded by bglu_2g07420 and composed of DUF2169 domain and pentapeptide repeats region, contributes to the bacterial virulence in rice plants. T6SS accessory protein having DUF2169 domain is well known as a T6SS adaptor. The DUF2169 domain-containing protein encoded by bglu_2g07420 could be considered as a potential T6SS adaptor, but additional experimental evidence, such as interaction with VgrG or specific T6SS-dependent effectors is required to confirm its function as a T6SS adaptor. (Line 324-330)”
For next paper, we are performing proteomics and the protein-protein interaction-based yeast-two hybrid system and co-immunoprecipitation as a follow-up study to confirm that the DUF2169 domain-containing protein encoded by bglu_2g07420 actually interacts with other T6SS components as well as rice proteins.

Round 2
Reviewer 2 Report
In figure 2, were WP_012733554.1 and WP_012733555.1 labeled right?
Author Response
We would like to thank you all for your time and valuable comments and are grateful for the opportunity provided to resubmit our revised manuscript entitled “T6SS accessory proteins, including DUF2169 domain-containing protein and pentapeptide repeats protein, contribute to bacterial virulence in T6SS group_5 of Burkholderia glumae BGR1” to Plants.
Please see below our specific response to the reviewers’ comments
The manuscript has undergone extensive modification of “English editing”. Also, the manuscript has been revised to improve the statement of research design (Line 112-114, Line 126-128).
To improve the conclusions supported by the results, we modified and revised the statement as :"In the current study, we obtained genes encoding the DUF2169 domain from the genome of B. glumae BGR1 and investigated whether the genes encoding the DUF2169 domain-containing proteins consisting of a single domain or multi-domain affect bacterial virulence to rice plants. Of the two genes encoding the DUF2169 domain-containing protein, bglu_1g23320 and bglu_2g07420, only the latter was involved in bacterial virulence (Line 327-331)." "Our results show that the DUF2169 domain-containing protein encoded in bglu_2g074720 is a multi-domain protein that requires both the DUF2169 domain and the pentapeptide repeats region to contribute to bacterial virulence (Line 334-335)."
Point 1: In figure 2, were WP_012733554.1 and WP_012733555.1 labeled right?
Response 1: We thank the reviewer for the valuable comments. WP_012733554.1 and WP_012733555.1 in Figure 1 have been modified to be displayed correctly.
We hope that the manuscript will now be eligible for publication.

Reviewer 3 Report
Thank you for addressing my comments. Nice paper.
Author Response
Dear the reviewers,
We would like to thank you all for your time and valuable comments and are grateful for the opportunity provided to resubmit our revised manuscript entitled “T6SS accessory proteins, including DUF2169 domain-containing protein and pentapeptide repeats protein, contribute to bacterial virulence in T6SS group_5 of Burkholderia glumae BGR1” to Plants.
The manuscript has undergone extensive modification of “English editing”. We hope that the manuscript will now be eligible for publication.

Reviewer 4 Report
The authors responded all the points to my comments. The ms seems to be improved for the publication.
Author Response

(The authors gave the same response as above.)
